# Lean Manual Assembly 4.0: A Systematic Review

**Adrian Miqueo [1,*]**, **Marta Torralba [2]** and **José A. Yagüe-Fabra [1]**

1   I3A-Universidad de Zaragoza, C/María de Luna 3, 50018 Zaragoza, Spain; jyague@unizar.es
2   Centro Universitario de la Defensa Zaragoza, Ctra. Huesca s/n, 50090 Zaragoza, Spain; martatg@unizar.es
*   Correspondence: adrian.miqueo@unizar.es; Tel.: +34-689-247-874

**Abstract:** In a demand context of mass customization, shifting towards the mass personalization of products, assembly operations face the trade-off between highly productive automated systems and flexible manual operators. Novel digital technologies—conceptualized as Industry 4.0—suggest the possibility of simultaneously achieving superior productivity and flexibility. This article aims to address how Industry 4.0 technologies could improve the productivity, flexibility and quality of assembly operations. A systematic literature review was carried out, including 234 peer-reviewed articles from 2010–2020. As a result, the analysis was structured addressing four sets of research questions regarding (1) assembly for mass customization; (2) Industry 4.0 and performance evaluation; (3) Lean production as a starting point for smart factories, and (4) the implications of Industry 4.0 for people in assembly operations. It was found that mass customization brings great complexity that needs to be addressed at different levels from a holistic point of view; that Industry 4.0 offers powerful tools to achieve superior productivity and flexibility in assembly; that Lean is a great starting point for implementing such changes; and that people need to be considered central to Assembly 4.0. Developing methodologies for implementing Industry 4.0 to achieve specific business goals remains an open research topic.

**Keywords:** assembly; lean; Industry 4.0; human-centered; operator 4.0

## 1. Introduction

The current situation of assembly operations is characterized by an increasingly varied demand (mass customization) while the production faces a trade-off between the superior productivity of automated assembly systems and the absolute flexibility and adaptablity of manual assembly. Therefore, high-volume production of discrete goods received heavy investments for automation, while low volume, made-to-order or engineer-to-order products were typically assembled manually [1,2]. In this context, Lean production (a generalization of the Toyota Production System) expanded from its origin—automotive—to many other sectors and was adapted as necessary to the particularities of each industry or company [3]. Lean production typically focuses on value as perceived from the customer's point of view. Thus it considers that the flexibility to quickly adapt to market demand is critical. For Lean, rigid automation can be seen as a hindrance rather than an advantage, and seeks to incorporate the human factor to automation: *jidoka*, or "automation with a human touch" [4].

The term Industry 4.0, initially adopted by a German strategic program [5], is used nowadays to express the relationship between different elements of the current manufacturing sector and the new digital technologies. These Key Enabling Technologies are, according to [6]: Big data and analytics, Autonomous robots, Simulation, Horizontal and vertical system integration, the Industrial Internet of Things (IoT), Cybersecurity, The cloud, Additive manufacturing, and Augmented Reality. Recent research on Industry 4.0 tends to focus on the possibilities brought by a certain new digital

technology or develops a framework to understand what would be the effect of incorporating such new technologies [7]. The arrival of the new digital technologies could address the aforementioned dichotomy of highly productive yet rigid automation vs. flexible but less-productive manual assembly. The quickly developing fields of human–robot collaboration, virtual/augmented reality and Automated quality control, to cite some examples, show promise in bringing forward actually flexible and adaptable automation that has the best of both worlds.

Scarcely explored is the development of implementation methodologies that bridge Industry 4.0 conceptual frameworks with the current state of industrial environments and the process to successfully deploy new digital technologies that bring the expected returns of investment. Additionally, if the Lean production approach and its techniques are also related to this implementation, the concept of Lean 4.0 could be used, as shown in the literature [8]. Since Lean production and Industry 4.0 certainly have some commonalities [9], Lean could prove useful in providing a starting point for the implementation of Industry 4.0 technologies that improve assembly operations in a mass customization demand context.

In order to assess the impact of any changes, careful evaluation systems are needed to ensure that technology investments are implemented to solve the problems and address business goals, and not just because they are available or they bring some cosmetic advantage.

The 4th Industrial Revolution is expected to transform the role of the people, but to what extent will assembly operators be affected—are humans to be replaced by machines or empowered by new technology?

The issue that this literature review aims to address is: *How could Industry 4.0 technologies improve the flexibility, productivity and quality of assembly operations?* To look into it, we aim to answer the following questions:

1. What are the characteristics and implications of mass customization for assembly operations?
2. What new Industry 4.0 digital technologies are relevant to assembly operations?How to make the most out of their potential, and how to measure the improvement?
3. Is Lean production the best starting ground for implementing Industry 4.0 assembly operations?
4. How would Industry 4.0 affect people in assembly? How to support people transitioning to Assembly 4.0?

To answer these questions, a systematic literature review was carried out. From these four sets of questions, six key concepts are extracted, as shown in Figure 1: The scope of this article is limited to *assembly* operations, particularly focusing on *mass customization* demand. Neither fully automated systems nor manual assembly deal comfortably with mass customization demand since one lacks flexibility and the other's productivity falls short. *Industry 4.0* aims to address this gap by providing superior connectivity between machines and people. *Lean production* may serve as a foundation for Assembly 4.0, transversally providing a framework to analyze and conceptualize the new role of *human operators*. Finally, to evaluate the efficiency of assembly systems, *Key Performance Indicators* are commonly used.

This article is structured in the following manner: Section 2—Materials and Methods—describes the methodology used for the review, which focuses on the six key concepts related to the issue being addressed. This section also includes a brief bibliometric analysis of the references used for the analysis. Section 3—Results—includes an analysis of literature, grouped into four main subsections: (3.1) Assembly operations, (3.2) Industry 4.0, (3.3) Lean, and (3.4) People. Each subsection focuses on one of the questions that this article aims to answer. Section 4—Discussion—gathers the main conclusions found in the previous analysis and addresses the main issue stated before.

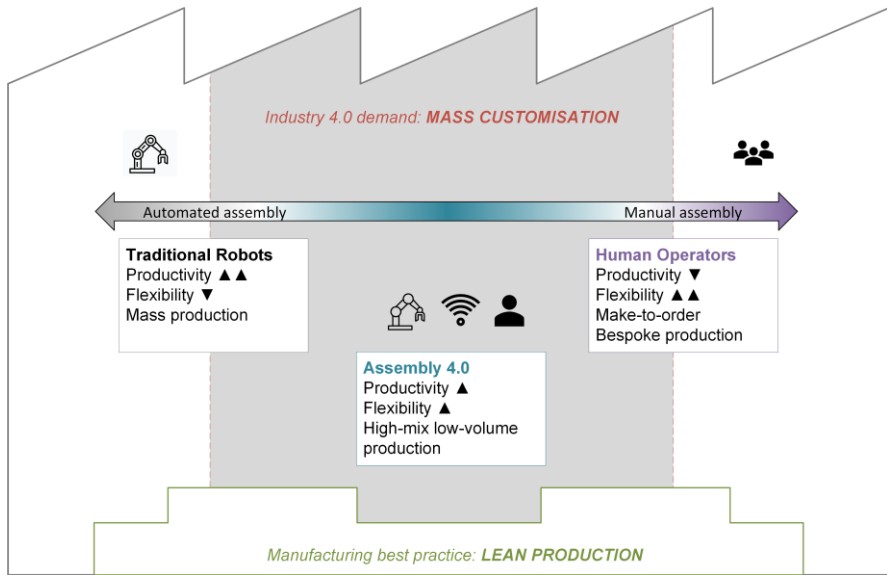

**Figure 1.** Key concepts used for the systematic literature review.

## 2. Materials and Methods

In order to address the issue introduced in the previous section and to answer the aforementioned questions, a systematic literature review was conducted. This section firstly describes the methodology employed in such a review, and secondly, offers a brief bibliometric analysis of the results.

The literature review was carried out in four stages—see Figure 2: database search, screening, eligibility and literature analysis.

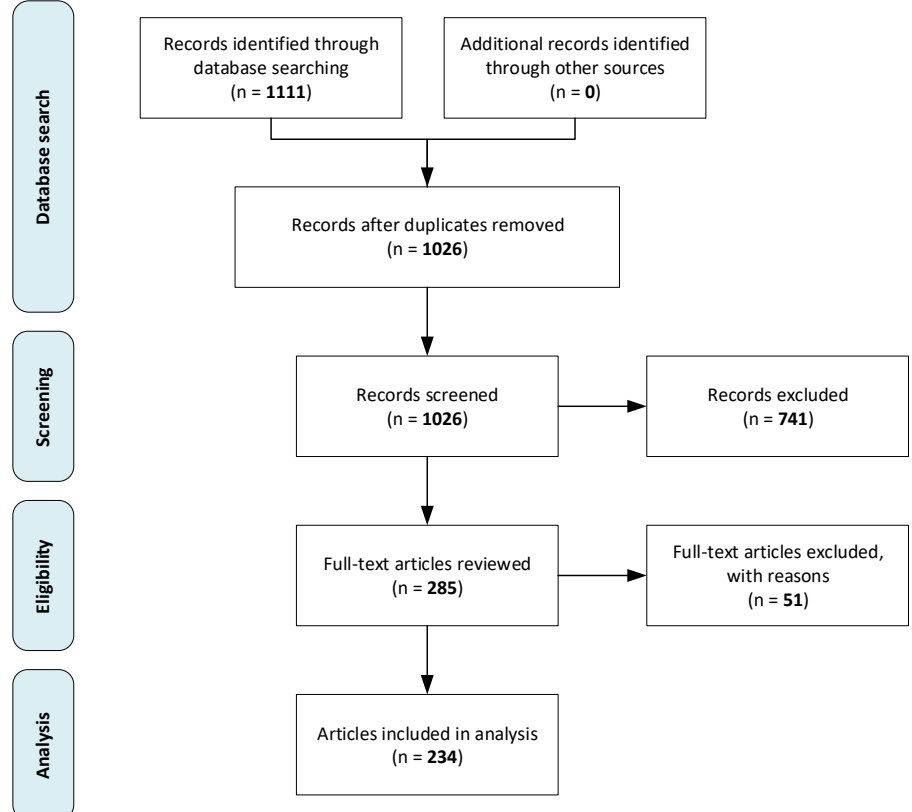

**Figure 2.** Search process and results, adapted from PRISMA [10].

The databases used for the initial stage were SCOPUS (Elsevier) and Web of Science, and only included relevant publications belonging to the following fields: Manufacturing engineering, Industrial engineering, Generalist engineering, Operations and management science. Since the topic under study is the conjunction of several broad subjects, we decided to conduct a systematic literature review that specifically targets their intersections. The six key concepts that were used are Assembly, Mass customization, Key Performance Indicator (KPI), Lean manufacturing, Industry 4.0 and Operator. These concepts were chosen for the search because they are the key ideas in the posed research questions–"Key Performance Indicators" being used for measuring improvement. The following keywords were used to perform the database search: (1) Lean: *Lean manufacturing*, *Lean production*; (2) Mass customization: *mass customization*, *mass customization*; (3) Industry 4.0: *Industry 4.0*, *Industrie 4.0*, *smart factories*; (4) KPI: *"KPI"*, *Key Performance Indicator*; (5) Assembly: *assembly*; (6) Operator: *operator*, *people*, *person*. The keywords were used for Title, Author Keyword and Keyword Plus (in WOS), except for KPI, which was also searched for in the Abstract field. From these six key concepts, 15 search groups were defined by intersecting each possible combination of two concepts, as shown in Table 1. Duplicates were removed at this point, resulting in 1026 publications identified.

**Table 1.** Search groups created by the intersection of each pair of key concepts and number of publications found.

| Search Group | Publications WOS | Publications SCOPUS | Publications Identified after Duplicates Removed |
|---|---|---|---|
| Assembly and mass customization | 58 | 52 | 97 |
| Assembly and KPI | 20 | 19 | 33 |
| Assembly and Lean | 81 | 106 | 168 |
| Assembly and Industry 4.0 | 47 | 10 | 55 |
| Assembly and operator | 83 | 196 | 268 |
| Industry 4.0 and Lean | 48 | 8 | 55 |
| Industry 4.0 and operator | 33 | 16 | 45 |
| Industry 4.0 and mass customization | 17 | 2 | 19 |
| Industry 4.0 and KPI | 11 | 2 | 12 |
| Lean and mass customization | 14 | 19 | 32 |
| Lean and KPI | 31 | 58 | 74 |
| Lean and operator | 10 | 33 | 40 |
| Operator and mass Mass Customisation | 4 | 15 | 15 |
| Operator and KPI | 13 | 98 | 108 |
| Mass customization and KPI | 4 | 3 | 5 |

The publications resulting from this search were then screened—based on title, abstract, publication and year—to assess which of them met the inclusion and exclusion criteria shown in Table 2, resulting in 741 records being excluded and 285 articles being included.

**Table 2.** Eligibility and exclusion criteria.

| Inclusion Criteria | Exclusion Criteria |
|---|---|
| Peer-reviewed publications | Book chapters |
| Recent: published in 2010 or later | Regarding construction, continuous production (e.g., petrochemical), energy efficiency |
| Language: publications in English | Regarding product design |
| | Regarding mathematical models or algorithms for scheduling, line sequencing, or line balancing |

Finally, the 285 articles were reviewed within each one of the 15 search groups and assessed for eligibility, resulting in 51 articles being excluded because they were not relevant to the key concept being analyzed.

The resulting 234 articles were analyzed, and the outcome of such analysis can be found in Section 3—Results.

The number of articles included in the analysis shows an increasing trend over time, as shown in Figure 3. It should be noted that the database search was performed in June 2020. Therefore the results shown in this analysis only include articles published up until the first half of 2020. It can be seen that the number of articles related to some key concepts remain constant or grow slightly over time—assembly, mass customization and operator—while others grow significantly—Lean and KPI. The number of articles related to Industry 4.0 is rising since 2015, which is consistent with the fact that the term "Industry 4.0" was coined in 2011 [5]. Of the 234 articles included in this review, 54 are conference or proceedings articles (23%), and 180 are journal articles (77%). The articles were published in a total of 117 publications, with 18 journals including 50% of the total articles and 83 publications contributing with just one article to this review. This is consistent with the database search strategy, which looks at the intersections of 6 different concepts.

**Figure 3.** Publications related to each key concept, by year.

## 3. Results

This section shows the outcome of the systematic literature review carried out following the methodology described in the previous section, and that addresses the issue of improving assembly operations in terms of productivity, flexibility and quality by using novel digital technologies of Industry 4.0. To look into this question, four specific questions were presented in the first section of this article. In consequence, this section is composed of four parts made of the search key concepts most closely related to each one of the questions, as shown in Figure 4. Firstly, looking into "the characteristics and implications of mass customization for assembly operations", the key concepts used are "assembly" and "mass customization" (3.1). Secondly, to identify "the new Industry 4.0 technologies, how to make the most out of them and how to measure the improvement", the key concepts used are "Industry 4.0′ and "Key Performance Indicators" (3.2). Then, the key concept "Lean" is employed to determine whether Lean production is the best starting ground for implementing the

aforementioned technologies (3.3). Finally, to explore "the effect of Industry 4.0 on people in assembly and to find out how to support them in transitioning to Assembly 4.0, the search key concept used is "operator" (3.4).

**Figure 4.** Research questions, search key concepts and their relationship to the literature review analysis topics.

### 3.1. Assembly Operations for Mass Customisation

In order to answer the first question, "What are the characteristics and implications of mass customization for assembly operations?" the systematic literature review publications related to the key concepts "assembly" and "mass customization" were analyzed. After a brief introduction, the five main topics to be considered will be presented, as shown in Figure 5: Modularity and product clustering; Mixed-model assembly optimization; Customer involvement and postponement strategies; The implications of complexity; and Mass customization impact on operators. Finally, the key conclusions will be summarized.

### 3.1.1. Introducing Assembly Operations for Mass Customisation

Mass customization demand is characterized by a combination of great variety, shorter product life cycles, and variable production volumes (medium or high for platform products, very low for personalized products); compared to Industry 2.0's stable market and Industry 3.0's volatile market—in terms of product volume, product variety and delivery time. In this new context, Toyota Production Systems (TPS) may prove limited, and its advantages and disadvantages with regards to *seru* were analyzed by Yin et al. [11]. The usage of new key digital technologies will bring forward the 4th Industrial Revolution (Industry 4.0), addressing many of the challenges of production systems for mass customization [11,12]. However, looking at isolated systems may not be enough since increased complexity requires a holistic approach to respond successfully and cost-effectively to shifting market demands [13]. Assembly is the final process to create a product, where component

sub-assemblies come together into the final product. Demand-driven increasing product variety adds complexity, production cost and lead time to assembly operations, which goes against its goals. In the mass customization landscape, key assembly topics need to be reviewed, evaluated and adapted [2]: assembly representation and sequencing, especially non-sequential assembly; assembly system design—considering line balancing, delayed product differentiation and performance evaluation; assembly system operations—with a focus on exploring reconfigurable assembly planning, mixed-model assembly scheduling, and dealing with complexity resulting from different sources; and the changing role of human operators.

**3.1 ASSEMBLY & MASS CUSTOMISATION**

*Q1: What are the characteristics and implications of mass customisation for assembly operations?*

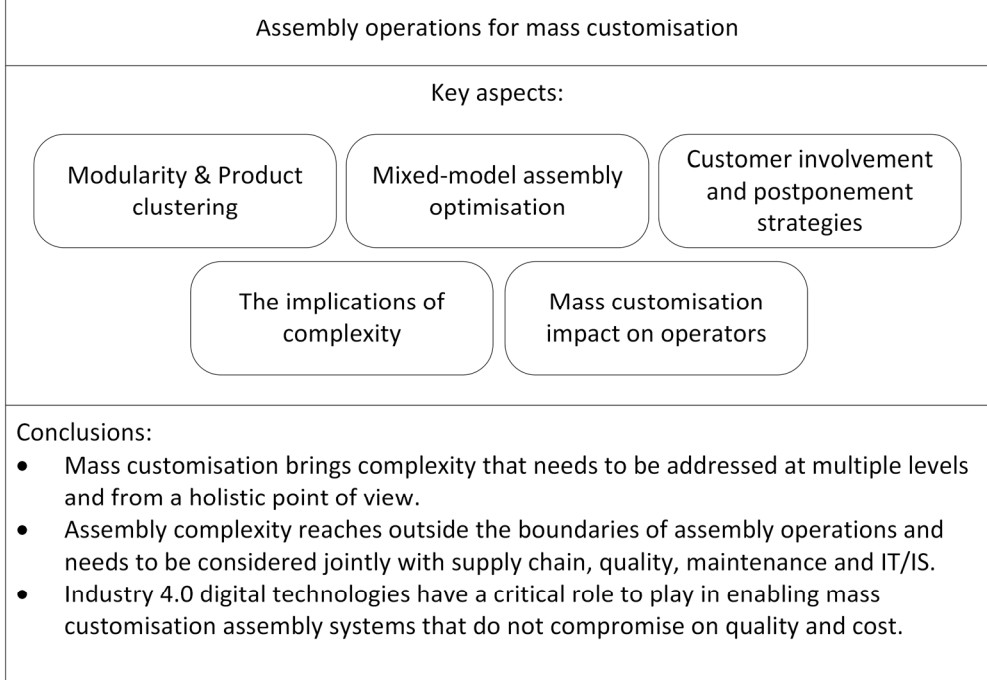

**Figure 5.** Key aspects of assembly operations for mass customization and main conclusions of the analysis.

**In conclusion**, mass customization brings increased complexity that needs to be addressed at multiple levels and taking a holistic point of view to ensure that optimizing a subsystem does not negatively affect another subsystem.

3.1.2. Modularity and Product Clustering

In order to flexibly assemble many different product variants using the same resources (such as people, equipment, management systems) to keep manufacturing costs down and productivity high. Efficient grouping of products into clusters or families is of paramount importance. The variables selected for clustering will depend on the assembly operation objectives, for instance: quality and cost to determine product family design [14]; product variety to determine assembly system layout [15]; assembly and disassembly for configuring product variants [16]; procedure, equipment and parts [17];

or involving worker's perspective for actual ease of assembly [18]. Modular production systems would also benefit from automated planning based on individual product CAD files [19].

**In conclusion**, product clustering, modularization, reconfigurable assembly systems and delayed product differentiation are valuable tools to maintain competitive assembly in a mass customization context.

### 3.1.3. Mixed-Model Assembly Optimisation

Another area greatly affecting the efficiency of assembly lines is its sequencing and balancing. Similar to clustering and modularization, different approaches are used depending on the focused goals of the optimization: cooperative sequencing or workstation analysis for assembly material consumption waviness, setup time and lead time [20,21]; multi-agent systems analysis for reducing the negative impact of material handling complexity [22]; monitoring manufacturing complexity for workload balancing [23]. New approaches have also been developed to optimize assembly line sequencing [24,25].

**In conclusion**, mixed-model assembly is needed to deal with mass customization while remaining competitive since it allows to address various operational goals depending on the business needs.

### 3.1.4. Customer Involvement and Postponement Strategies

Mass customization may be leading towards mass personalization, where individual products made to match the exact preferences of each customer are produced in large numbers [1]. Integrating the customer in the design phase could be done using web-based platforms [26], while Industry 4.0's Cyber Physical Systems (CPS) and a tailored assembly architecture would enable efficient mass personalization [27]. An alternative strategy is Postponement, which could help with dealing with high assembly complexity [28]. However, it requires designing the assembly line layout for delayed product differentiation [29,30] and would benefit from reconfigurable assembly stations [31].

**In conclusion**, assembly operations need to consider the increasing expectations of mass customization heading towards mass personalization. In order to adapt to it, Industry 4.0 Cyber Physical Systems could be used to develop reconfigurable assembly stations that can deal with high assembly complexity while maintaining high productivity.

### 3.1.5. The Implications of Complexity

Mass customization brings a great deal of complexity to assembly operations, which affect key elements of the system as well as other nearby areas, such as quality, supply chain or maintenance.Assembly complexity has can been evaluated from different perspectives: the number of product variants [32], induced task differences [33] or product configuration [34]. Complexity has a negative effect on quality, which could be minimized by using cognitive automation [35]. The increasing number of product features to be controlled makes the necessary new advanced quality management systems [36]. Supply chain implications of mass customization assembly range from assembly line feeding problems [37] and modularity-specific issues [38] to assembly supply chain configuration [39] and whole manufacturing networks [40]. Using Automated Guided Vehicles (AGVs) can be used efficiently to feed mixed-model assembly lines [41,42]. Maintenance resource allocation also needs to be prioritized to minimize the negative effects of increased complexity [43].

**In conclusion**, assembly complexity reaches outside the boundaries of assembly operations and needs to be considered jointly with supply chain, quality, maintenance and IT/IS.

### 3.1.6. Mass Customisation Impacts Operators

Fully automated assembly systems bring increased productivity for high-volume production but lack the flexibility and adaptability of human operators. People are better equipped for assembly tasks with small and frequent variations, but their potential for higher productivity is limited. In a context of market demand characterized by mass customization, which heads towards mass personalized

production, reconfigurable assembly systems that incorporate both machines and people can lead to cost-effective systems that are flexible and scalable [2]. Automation needs to consider both the physical and cognitive abilities of the human operators it supports [44].

In order to improve the yield of assembly operations, providing support to human workers is necessary. Augmented Reality (AR) could be used, reducing the number of engineering/production management resources needed to provide assembly operators with cognitive support to perform their tasks [45,46]; as well as cognitive/handling skills transfer systems [47], self-adapting automatic quality control [48] or cognitive automation strategies [49]. Automation needs to ensure human safety, which led to research on Human–Robot Collaboration (HRC) plan recognition and trajectory prediction [50], and the concept of "safety bubble" [51]. When employing novel digital technologies for enhancing assembly systems performance, one cannot underestimate the strategic importance of IT/IS systems [52].

**In conclusion**, in a context of market demand characterized by mass customization, which heads towards mass personalized production, reconfigurable assembly systems that incorporate both machines and people can lead to cost-effective systems that are flexible and scalable. Industry 4.0 digital technologies have a critical role to play in making possible mass customization assembly systems that do not compromise on quality and cost and that do not achieve increased performance by affecting human operators negatively.

### 3.1.7. Assembly and Mass Customisation: Conclusions

In a context of market demand characterized by mass customization which heads towards mass personalized production, the increased complexity reaches the boundaries of assembly operations and needs to be considered jointly with other areas (e.g., supply chain, quality, maintenance, IT/IS) and taking a holistic point of view to ensure that optimizing a subsystem does not affect others negatively. To maintain assembly operations competitive despite the increased complexity, product clustering, modularization, delayed product differentiation, mixed-model assembly, and reconfigurable assembly systems are valuable tools. Reconfigurable assembly systems in which human operators work effectively alongside machines or robots, made possible with Cyber Physical Systems, can lead to cost-effective systems that are flexible and scalable. It seems clear that Industry 4.0 digital technologies have a critical role to play in making possible mass customization assembly systems.

### 3.2. New Digital Technology Available: Industry 4.0

In order to answer the previously presented questions "What new Industry 4.0 digital technologies are relevant to assembly operations?", "How to make the most out of them?" and "How to measure the improvement?"; the systematic literature review publications related to the key concepts "Industry 4.0′ and "Key Performance Indicators" were analyzed. After a brief introduction on Industry 4.0 (I4.0), the eight main topics to be considered are presented, as shown in Figure 6: I4.0 technology for improving processes and decisions; I4.0 technology for mass customization; I4.0 technology for supporting human operators; I4.0 for mass customization; Key Performance Indicators for assembly; Key Performance Indicators for I4.0; and Small and Medium Enterprises (SMEs) in the I4.0 era. Finally, the key conclusions are summarized.

### 3.2.1. Introducing "Assembly 4.0"

According to Yin et al., industrial revolutions are related to distinct technologies, market demands and production systems. The 4th industrial revolution differs from industry 1.0–3.0 because it is expected to happen in the near future, as opposed to the previous three. The deep and intertwined changes in available technology and market demand paradigms create new possibilities; however, the industry 4.0 production systems are expected to be an evolution from the previously existing systems (characterized by *seru*, flow lines, Toyota Production System or TPS, job shops, cellular manufacturing and Flexible Manufacturing Systems or FMS) enhanced by the novel digital technologies [11].

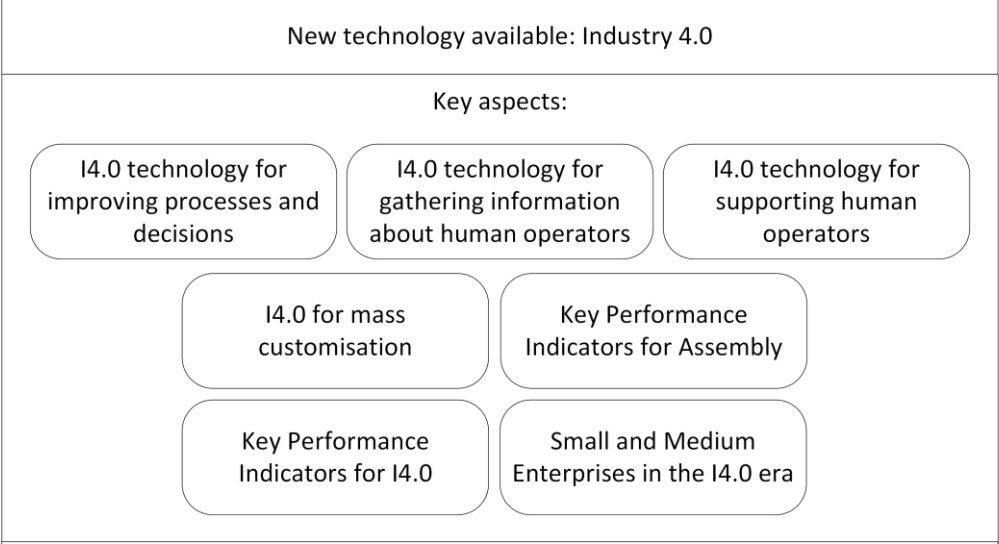

**Figure 6.** Key aspects of Industry 4.0 technologies for assembly operations and Key Performance Indicators (KPIs), and main conclusions of the analysis.

Bortolini et al. investigated in [53] the impact of the 4th industrial revolution on assembly systems design. The dimensions to consider are six: balancing, sequencing, material feeding, ergonomic risk, equipment selection and learning effect. The evolution of the industrial environment in European countries leads to an aging workforce, re-shoring of production facilities and more efficient and distributed communication networks. In this environment, nine are the enabling technologies of Industry 4.0 that have the most potential to affect assembly systems: big data, IoT, real-time optimization, cloud computing, Cyber Physical Systems, machine learning, Augmented Reality, cobots and additive manufacturing. The integration of these technologies in the design and management of assembly processes leads to what Bortolini et al. define as "*AS40*": Assembly Systems 4.0. The main

characteristics of *AS40* are assembly control systems, aided assembly, intelligent storage management, late customization, product and process traceability and self-configured workstation layout [53].

Cohen et al. looked into how assembly system configuration would be affected by Industry 4.0 principles, understood as four incremental stages or steps to achieve the 4th revolution: connectivity, information, knowledge and *smart,* which involves "predictive and automated decision making processes, with possible self-adjustments and reconfiguration of the production system". The new paradigm would reduce the costs of assembly automation; reduce setup costs and learning curves; enable the assembly of small quantities of large products in flow lines; enable the assembly of very different products in the same system; better traceability of failures and defects; and smarter material handling. In the last stage of Industry 4.0 (*smart*), assembly systems would be Self-Adapting Smart Systems (SASS), and together with continuous support to operators (OSS), flexibility, agility and productivity would be greatly increased [54].

According to Cohen et al. in [7], the main goal of flexible assembly systems in the Industry 4.0 era is to address the mass customization demand paradigm. At this moment, operational, tactical and strategical issues remain unsolved for implementing "Assembly 4.0". A key aspect is the social effect of Assembly 4.0: the assembly workforce is expected to shrink—at least, in Western countries-, but additional technological job positions will appear, partially offsetting the operator reduction. The workforce would experiment a net decrease, thus increasing the productivity per employee. Therefore, the role of people in A4.0 will be increasingly important, which calls for future research that considers human operators back at the center of the production systems of the future [7].

When looking ahead in the evolution of assembly systems into the 4th Industrial revolution, Cohen et al. identify challenges when integrating new and existing technologies: uncertainty on the synergies of the I4.0 Key Enabling Technologies; the human–automation collaboration; incorporating Artificial Intelligence into assembly systems; and finding, developing and keeping the Assembly 4.0 human specialists. On top of the technical knowledge, Industry 4.0 operators will need a new set of non-technical skills, so education centers and companies will need to work together to meet this demand [55].

Developing an Assembly 4.0 system in a controlled environment, such as a Learning Factory, allows to better understand the complexity of such a system. The drone factory developed by Fast-Berglund et al. "focuses on the interaction and cooperation between humans and cobots to create collaborative applications in final assembly tasks". It was built with operator involvement from the start, and it incorporates a modular and event-driven IT architecture that creates a digital twin of both product and production system, allowing automated planning and preparation of operations [56].

Facing a mass customization demand, late customization is a strategy allowing customers to make changes to their orders even when the production has started. Industry 4.0 digital technologies bring additional tools for developing an assembly system able to cope with resequencing the production process [24]. Identifying information and data needs is a key step in the design of smart assembly factories to ensure that the increased complexity associated with addressing mass customization production can be managed by human operators [57]. Additionally, strategies for improving the use of IT/IS systems in assembly need to consider the whole digital strategy of the organization [52]. Optimizing the design of any Industry 4.0-enabled system at early stages is critical for SMEs in the manufacturing sector. Axiomatic design and *Acclaro* software have proven useful [58].

The analysis of literature allowed to organize Industry 4.0 technologies in four main categories depending on their goals in assembly operations: improving processes and decisions, gathering information on human operators, supporting people in assembly, and enabling mass customization. Table 3 summarizes the references to technologies employed for each goal.

### 3.2.2. Industry 4.0 Technologies for Improving Processes and Decisions

Novel Industry 4.0 technologies can be used to improve processes and gather meaningful data, which allows better-informed decisions. Big data can be used to maximize yield and machine uptime in

precision assembly processes by detecting long term errors and enabling predictive maintenance [59]. Sensors from across the shop-floor can be used in conjunction with an IT/IS service to provide critical information about the processes in the white goods industry [61]. RFID can be used to track assembly execution and then to derive guidelines for smart assembly line development [62] and web-based systems (*saas*) to control smart internal logistics using mobile robots [68]. Motion Analysis System (MAS) to monitor and evaluate manual production processes [63,93]. The *Human Factor Analyzer* is a software/hardware architecture that can be used for manual work motion and time measurement employing depth cameras and automatic data processing aiming to evaluate work performance quantitatively [64]. Digital twins of assembly processes can be used to analyze the efficiency of the line [85], and it would also enable product-centric assembly [86]. Festo's *Cyber Physical Factory* can be used to implement an Industry 4.0 digital twin framework [87].

**Table 3.** Technologies of Industry 4.0 by usage.

| Industry 4.0 Technologies [1] | Improving Processes and Decisions | Gathering Information on Human Operators | Supporting People in Assembly | Enabling Mass Customisation |
|---|---|---|---|---|
| Big data | [59] | | [60] | |
| IoT | [61–64] | [65] | [66] | [67] |
| Real-time optimization | [68] | [69] | | [24] |
| Cloud computing | | | | [70] |
| Cyber Physical Systems | | | [71,72] | [67] |
| Augmented/Virtual Reality | | | [73–82] | [83] |
| Additive manufacturing | | | | [83,84] |
| digital twin | [85–87] | [69] | | |
| Other | | [88,89] | [49,90–92] | |

[1] Industry 4.0 Key Enabling Technologies based on [53].

### 3.2.3. Industry 4.0 Technologies for Gathering Information on Human Operators

Industry 4.0 technologies allow new ways of gathering information about human assembly operators that are less intrusive, more accurate or more capable than previously existing techniques: Mattson et al. propose a method of measuring the wellbeing and performance of operators at assembly stations [88]. Krugh et al. measure human–machine interaction using the Internet of Things (IoT) to understand the impact of people on Industry 4.0 assembly systems [65]. Eye-tracking can be used to analyze the user experience of engineering design and manufacturing [89]. A theoretical human-centered framework for operator 4.0 using digital twin based simulation and real-time human data capture can be used to provide insights on operator ergonomics and mental workload [69].

### 3.2.4. Industry 4.0 Technologies for Supporting People in Assembly

Cyber Physical Systems (CPS) for improving operator ergonomics [71]; vision systems for measuring and providing feedback on operator performance [90]; cognitive assistance for rework area [91]; strategies for cognitive automation that allow operators to deal with increased complexity [49]; Augmented Reality (AR) to assist manual assembly [73]; operator training using digital assistance [92]; training using Virtual Reality and process mining, allowing to replace traditional interpersonal demonstration and repetition [74] and real-time interface using data from many devices and an algorithm allowing manual assembly operators to deal with requests and report faults [66].

### 3.2.5. Industry 4.0 Technologies for Mass Customisation

Manufacturing flexibility is a strategic orientation for high-wage countries, and Industry 4.0 technologies bring solid benefits to operations management, especially in terms of technology management and Just-In-Time (JIT) production [94]. One technology in particular—additive manufacturing, can break the flexibility vs. cost trade-off, which most industrially developed countries face [84]. Compared to the volatile market of Industry 3.0, characterized by product variety, the smart

market of industry 4.0 involves customer participation in individual customization of products [11]. Industry 4.0 KET enable mass personalization through short product development cycles [83] and individual customers' input [67,95]. Rossit et al. propose an approach based on tolerance planning strategies and resequencing capabilities to allow changes to the product to be made even after production has started [24]; while Chung et al. envisage a dynamic supply chain design for connected factories through cloud-based information systems as a way to achieve mass personalization [70].

**In conclusion**, Industry 4.0 not only offers new alternatives for cost-competitive mass customization but also opens the door to mass personalization, where the customer is involved in individual customization of the product.

### 3.2.6. Key Performance Indicators for Assembly

Key Performance Indicators (KPIs) are employed widely to assess the outcome of assembly systems. New concepts for novel assembly systems need to use KPIs to evaluate their potential performance. In most cases, traditional KPIs are used [96]: cost (investment, labor), quality (first pass yield, final yield) [97–99], throughput time, quantity and lot size; inventory costs [100], line productivity (e.g., OEE—overall equipment effectiveness) [101], energy consumption, cycle time and service level [102,103]. Integrating KPIs that link design, production, and quality goals through the product & process development has proven useful to limit late engineering changes, which delay the assembly system development [104]. A combination of economic and structural KPIs can be used to evaluate the adaptability of reconfigurable manufacturing systems [105]. Yang et al. propose that KPI selection for the smart automation of manufacturing systems needs to be company and location-specific and that the KPIs variation and sensitivity to the introduction of new Industry 4.0 technology needs to be a key driver for developing a strategy for smart assembly automation [106]. For evaluating the performance of Line-less Mobile Assembly Systems (LMAS), Hüttemann et al. developed a set of 11 specific KPIs, 6 of which are adapted from conventional KPIs to account for the wide variety of products being made in the assembly system, and 5 are specific to LMAS (e.g., overall traveled distance, number of station configuration reconfigurations) [107].

**In conclusion**, to evaluate assembly systems, standard KPIs need to be adapted in order to include both traditional metrics (e.g., cost, quality, throughput, inventory, lead time, productivity) and new indicators that are specific to the products, operations context and business goals.

### 3.2.7. Key Performance Indicators for Industry 4.0

Manufacturing flexibility is a strategic orientation for high-wage countries, and Industry 4.0 Key Enabling Performance measurement is a necessary management tool in any factory transformation. Traditional KPIs are valid to evaluate the impact of Industry 4.0 on production systems. However, new IT-related KPI classes will be required to assess data management (e.g., IT efficiency, availability of IT, the correctness of data, completeness of data), transparency & connectivity (e.g., degree of interconnectivity, digital coverage, the proportion of virtually controllable resources), and product management [108]. Industry 4.0 technologies bring the possibility of using IoT devices to gather real-time data from an immense number of devices in real time, enabling rapid responses to changing conditions [109]. KPIs for smart factories need to be reliable and targeting the right goals to support operational objectives. Therefore, correctly identifying the smart factory stakeholders and understanding their requirements is crucial [110]. Transforming a traditional factory—using legacy machines—into a smart factory is possible without buying expensive new machines, employing a continuous improvement approach, the IoT as enabling technology and establishing visible KPIs from the beginning so that the path to Industry 4.0 is clear to all stakeholders [111]. The increased network complexity and data traffic increase the probabilities of IoT failure. To address this, a data anomaly response model was proposed by Hwang et al. [112]. The changes brought by Industry 4.0 could affect people greatly. To make this impact on people more visible, human-centric KPIs have been proposed [113].

**In conclusion**, traditional and new IT-related KPIs classes (e.g., data management, transparency and connectivity, product management) would be used to assess and control the impact of Industry 4.0 on production systems. Identifying the smart factory stakeholders and their requirements is critical for obtaining meaningful KPIs. The Internet of things is the Key Enabling Technology that allows gathering data from multiple sources to produce real-time KPIs that allow rapid responses to fast changes in smart factories.

### 3.2.8. Small and Medium Enterprises in the Industry 4.0 Era

Although large corporations are more likely to benefit from adopting Industry 4.0 technologies, Small and Medium Enterprises (SMEs) could also obtain a competitive edge from Lean-digital manufacturing systems [114]; for example, improving the communication between shop-floor and the top-floor [115]. SMEs have different needs and requirements, which should be taken into account when designing smart manufacturing systems [116]. SMEs have started their digitalization journey, but further Industry 4.0 developments need to align with the particularities of SMEs, and their organizational structures need to fully embrace and support digitalization in order to benefit from its implementation [117]. Fast-Berglund et al. looked at 40 SME and 8 OEMs in order to establish collaborative robot (cobots) implementation strategies and to determine what KPIs to use for these cases [118]. The increasing penetration of intelligent machines to work alongside people and the benefits of *agile* production will turn SME operators into "Makers", skilled workers whose main activities are no longer assisting or monitoring machines, but creative tasks involving a wealth of information, alternatives, criteria and possible solutions [119].

**In conclusion**, Small and Medium Enterprises (SMEs) operators will be affected differently by I4.0 compared to corporate workers, but it is clear that I4.0 can bring competitive benefits for SMEs.

### 3.2.9. Assembly 4.0: Conclusions

The 4th Industrial revolution demand paradigm means mass customization of products, made possible by new digital technology. Conversely, production systems are most likely to experiment an evolution rather than a revolutionary change. Two key areas will be subject to change: the role of people in assembly operations—especially in terms of responsibility and skills; and the possibility of automated or hybrid assembly for low-volume production, including multi-mixed model assembly.

To evaluate the performance of assembly systems, standard KPIs need to be adapted in order to include both traditional metrics (e.g., cost, quality, throughput, inventory, lead time, productivity) and new indicators that are specific to the products, operations, stakeholders and business goals. The Internet of Things is the Key Enabling Technology that allows gathering data from multiple sources to produce real-time KPIs that allow rapid responses to fast changes in smart factories. The smart factory will need to consider also IT-related KPIs to ensure its smooth computer-dependent operations.

There are plenty of examples of new possibilities due to novel technologies applied to final assembly: improving processes, gathering data and obtaining valuable information, measuring human operator performance and supporting human operators" work. However, research articles mostly focus on what the new technology can do, but few relate to following a methodology to assess the operational needs or opportunities in final assembly and finding or developing an Industry 4.0 solution to them.

In order to ensure that the solutions enabled by Industry 4.0 technologies are aimed in the right direction, it is important to keep the focus on adding value.

### 3.3. Focusing on Delivering Value: Lean

In order to answer the third question, "Is Lean production the best starting ground for implementing Industry 4.0 assembly operations?" a systematic literature review of publications related to the key concept "Lean" was analyzed. After a brief introduction, the nine main topics to be considered are presented, as shown in Figure 7: Lean tools for assembly operations; Internal logistics; Ergonomics;

Assembly operations layout; Teaching Lean; Evaluating performance; Lean and Industry 4.0 interaction; Lean tools for Industry 4.0; and Lean management. Finally, the key conclusions are summarized.

<div style="background-color:#d6e0c2; padding:10px; text-align:center;">

**3.3. LEAN**

</div>

*Q3: Is Lean Production the best starting ground for implementing Industry 4.0 assembly operations?*

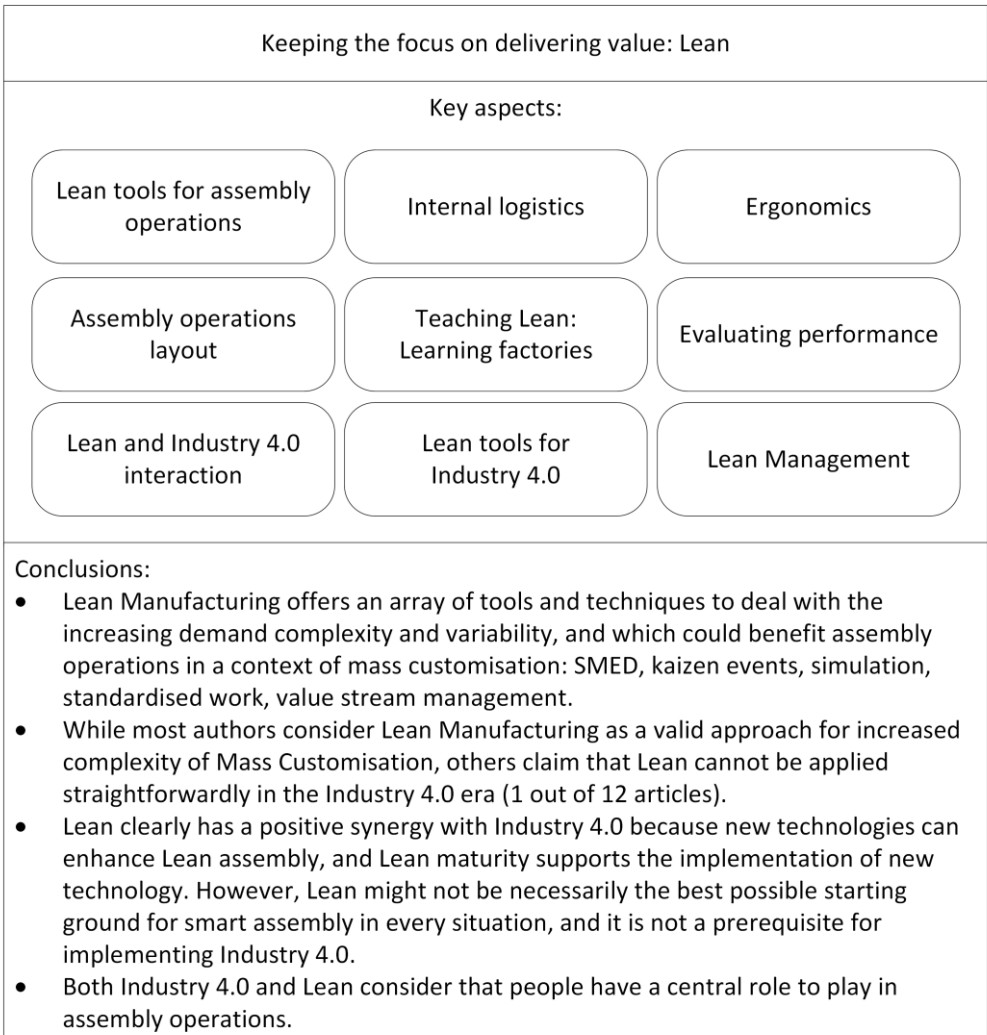

**Figure 7.** Key aspects of Lean assembly for Industry 4.0, and the main conclusions of the analysis.

### 3.3.1. Introducing Lean in the Era of Industry 4.0

According to Yin et al., one key characteristic of the Industry 3.0 market—product variety—changed is to change in the Industry 4.0 era to mass customization (customer participate in individual customization). However, the existing production systems will not change in a great way, as flow lines, Lean production, cells, and remain up to date when facing mass customization [11]. On the other hand, Stump et al. propose that despite the fact that Lean production can be applied easily to manufacturing situations with low levels of customization (i.e., product variety, Yin's Industry 3.0 market conditions), increasing levels of customization make it difficult to directly apply Lean principles of establishing flow and keeping low inventory levels [120].

Gunasekaran et al.'s review conclude that Agile manufacturing (which shares with Lean its focus on product value as defined by the customer) is key for sustainable competitive advantages; and identifies five enabling competencies that need to be deployed jointly to achieve its goals: transparent customization, agile supply chains, intelligent automation, total employee empowerment and technology integration [121]. To cope with mass customization with Lean objectives of continuous mixed-model flow, Chatzopoulos presented a production system design algorithm that employs production modules connected by Kanban [122].

### 3.3.2. Lean Production Tools for Assembly Operations

Lean manufacturing offers an array of tools and techniques to deal with increasing demand complexity and variability which could benefit assembly operations in the context of mass customization. Although Lean, a generalization of the Toyota Production System (TPS), originated in the automotive industry, it has expanded to many other manufacturing sectors—e.g., aeronautical, which demand characteristics are not similar to automotive [123]. One classic Lean tool is a single minute exchange of die (SMED), which is still a trending topic according to a recent review [124]. Looking at balancing manual assembly lines with a high number of product variants (mixed-model assembly), kaizen events and complexity reduction have proven useful since they fill the gap between mathematical balancing models developed by academia and actual techniques used in industry [125]. Mixed-model assembly lines throughput rate can be increased by using Lean in conjunction with simulation [126]. To increase productivity and reduce the necessary shop floor space, continuous flow can be achieved through the use of Standardized Work (SW), U-shape assembly lines and material handling systems [127]. Continuous improvement tools can be applied to increase throughput and reduce buffer capacity [128]. To address the increasing complexity of SW for mixed-model assembly, a reconfigurable approach to SW sheets and control and fabrication instructions has proven useful [129]. Value Stream Mapping (VSM), another classic Lean tool, has been evolved into value stream management at the University of Luxembourg Lean Manufacturing Laboratory [130]. A different approach to VSM is combining electronic-VSM with simulation, resulting in reduced lead times and non-value-added activities [131]. Three new methods were proposed to identify non-obvious constraints of mature production processes, where traditional Theory of Constraints methods fall short [132].

**In conclusion**, research on the application of Lean techniques and tools for assembly operations is still an open topic. The digitalization of some of the tools, such as Value Stream Mapping, has shown some success.

### 3.3.3. Internal Logistics

An adjacent key area to Lean assembly operations is Logistics, which makes the necessary components or materials available for assembly at the right time with minimum waste. Lean supply chain uses six classic KPIs: lead time, costs, inventory level, delivery service level and quality [100]. To increase the assembly line's value-add time and ergonomics, and to reduce waste and necessary space, using plastic containers instead of cardboard has been found an interesting option [133]. Looking into minimizing Work-In-Progress stock (WIP) and the required number of assembly operators, pre-kitting offers advantages as well as challenges [134,135]. Usta et al. propose a methodology for assessing the best design for part feeding system for Lean assembly, considering that the problems of pure kitting could be countered by hybrid systems (human & machine) [136]. Yamazaki et al. present a design method to reduce the cost of flexible automation of material handling systems [137]. In-house logistics for Lean assembly require evaluating and selecting from different transportation alternatives in order to feed part supermarkets [138].

**In conclusion**, internal logistics are tightly associated with assembly, and therefore both should be analyzed together since changes to one will affect the other as well.

### 3.3.4. Ergonomics

Lean production (LP) impact on ergonomics and psychosocial risks have been studied for decades, and the focus of the studies has varied over time, with a current view that considers that management style can make LP effects either negative or positive [139]. Da Silva et al. develop an index to assess the LP assembly cell work in terms of ergonomics and psychophysical demand [140]. The impact of line and assembly cells on breaks and worker's health has been assessed, finding that assembly cells tend to have higher Cycle Times, which increase the physicality of the work; while assembly lines posed no risks [141]. A different approach to evaluating the impact of LP on ergonomics is utilizing simulation: (1) for analyzing the effect of physical overload on assembly line performance, finding that Cycle Times too close to TAKT (i.e., low catch back time) leads to operator overload, which means absenteeism and low productivity in the long term [142]; (2) or for designing efficient hybrid assembly lines that are ergonomically safe [143].

**In conclusion**, Lean production can affect ergonomics negatively depending on management style.

### 3.3.5. Assembly Operations Layout

A key aspect of Lean assembly operations is the production layout. Classic Lean assembly is done in assembly lines or assembly cells. Assembly cells offer various advantages with regards to assembly lines, and a methodology for reconfiguring an assembly line into a cell is proposed by Carmo-Silva et al. [144]. The efficiency of Lean manufacturing production systems can be better analyzed when considering assembly as a macro-activity instead of a series of stations, and the identification of the waste is fine-tuned to assembly operations [145]. Lean assembly lines typically use Kanban to pull production and create material flow. In his paper, Savino et al. propose a method for using semi-automated parts feeding in O-shaped assembly lines [146].

Yin et al. analyzed in [147] the similarities and differences between Lean assembly (lines and cells), Agile manufacturing (Quick Response Manufacturing, QRM) and *seru* manufacturing. They found, based on two key industrial cases (Canon and Sony), that a production system that focuses primarily on responding to quick changes in demand and product instead of prioritizing waste reduction (i.e., Lean production) can be very competitive in high-cost environments. As a result, of this priority, *seru* focuses on "reconfigurability, resource completeness within cells, worker responsibility and buffering as needed to accommodate dimensions of demand variability". However, the applicability of *seru* assembly systems outside of high-cost, high variability, high innovation, short product development cycles remains to be seen [147].

**In conclusion**, Lean production systems typically employ assembly lines or cell layouts to establish pull and create material flow. For certain contexts involving high-cost, high-variability, short product development cycles, *seru* assembly systems are particularly competitive because they are focused on adaptability.

### 3.3.6. Teaching Lean for Assembly Operations: Learning Factories

Since operator engagement is at the core of Lean production, Lean-assembly-focused training has been explored over the past decades. Academia-driven teaching methods have not always been adequately adapted for non-students. Recreating industrially relevant environments for teaching Lean at Learning Factories aim to bridge this gap [148]. Lean techniques themselves have been used to design a Learning Factory, using a manual assembly line as a starting point, and employing theoretical knowledge as well as industrial experience for evolving the line into a Learning Factory [149].

Learning factories are incorporating Industry 4.0 technologies into their education and research facilities, focusing on dealing with complexity [150], intelligent logistics [151] or intelligent manufacturing in full-scale simulations [152]. Virtual Reality (VR) and Augmented Reality (AR) can be used to enhance the student's experience when learning Lean manufacturing. Using VR for training and AR for visualizing the assembly instructions improved the lessons [153].

**In conclusion**, Lean Learning Factories need to mimic real-life scenarios to become useful for non-academic learners with industrial backgrounds, such as assembly operators. Industry 4.0 technologies could be used to enhance the training environment of Learning Factories.

### 3.3.7. Evaluating Performance From a Lean Perspective

Lanza et al. propose a simulation-based method for assessing the performance improvement of production systems due to Lean techniques. As Key Performance Indicators (KPIs), either direct measures or monetary equivalents are used to compare initial vs. future scenarios. To relate cost-savings over time, cost–time profile charts can be employed [154]. Complex coefficient KPIs derived from delivery date and balanced production can be used to assess small-batch mixed-model scheduling models better than simple KPIs, although the potential use of such KPIs to manage real operations is reduced [25]. Multi-criteria KPIs can be used not only for management and control of operations but at earlier stages of flow planning projects [155]. For practical results, leading indicators are preferred over lagging KPIs [156], so Cyber Physical Systems (CPS), which lead to intra-logistics evaluation tools that use a wealth of data collected automatically, could be preferred over-relying on human input [157].

Evaluating the operational performance of Lean organizations can be done using tree-like KPI structures [158] or integrated performance assessment frameworks [159,160]. Cortes et al. proposed a "Lean & Six Sigma Framework" [161] to evaluate leanness in order to justify future investment—in a similar fashion to Lanza et al.'s [154]—and focus on a methodology for a solid KPI definition that allows and enables strategic-operational alignment. Kovacs et al. studied the relationship between Lean maturity, operational performance and investment; and concluded that implementing and sustaining Lean practices pays off because new technology cannot improve performance if the processes are not under control in the first place [162].

**In conclusion**, KPIs and performance assessment frameworks are used to measure the effects of changes in Lean production systems. Establishing a set of KPIs needs to take into account multiple stakeholders and to align the strategic and operational goals of the organization. Simulations and case studies show the beneficial effects of Lean methods and allow to estimate the economic return of investment of Lean management decisions.

### 3.3.8. The Interaction between Lean Production and Industry 4.0

Lean production is a key characteristic of the 3rd industrial revolution production systems. While other aspects have evolved (e.g., technology, from computers to smart digital devices) or radically changed (e.g., market focus from variety and lead time to customization and personalization), Lean is still up-to-date in the era of Industry 4.0 [11]. Moreover, the relationship between Lean and Industry 4.0 technologies is catching increasing attention from academia in the last decade [163].

The question posed by Mrugalska et al. [164] has been addressed by many authors, both theoretically and analyzing use cases across many countries: "Can Lean and Industry 4.0 coexist and support each other, and if so, how?" There are four main lines of thought when answering this question: (1) Lean techniques and Industry 4.0 technologies interact in a positive way, and there are many cases to illustrate this [8,9,165,166]; (2) Lean facilitates the change towards Industry 4.0 [167,168]; (3) Industry 4.0 supports Lean, i.e., makes the factory Lean [169–173]; (4) although Lean and Industry 4.0 aim for the same goals, their approach is essentially different regarding digital technology [174].

Five articles looked at answering Mrugalska et al.'s question [164] by surveying the industrial reality of different countries, all of them finding positive interactions between Lean and Industry 4.0 technologies. Dombrowski et al. analyzed 260 industrial companies in Germany and found Lean as an enabler of Industry 4.0 [168]. Tortorella et al. looked into 110 user cases in Brazil and found a positive Lean-Industry 4.0 correlation, as well as increased benefits of new digital technologies where Lean was also present [175]. Rossini et al. analyzed 108 cases of European manufacturers, concluding that Lean allows achieving higher levels of Industry 4.0 while lacking Lean production techniques makes it more difficult to change towards Industry 4.0 [176]. Chiarini et al. investigated 200 cases in Italy and found

that most strategic, operational areas benefit from implementing Industry 4.0, such as design-to-cost, supply chain integration or machinery–electronics–database integration [177]. Lorenz et al. analyzed user cases in Switzerland and found that Lean maturity allows greater performance improvements from implementing Industry 4.0 [178].

In conclusion, there is a wealth of evidence showing that Lean manufacturing is a valid approach to improve assembly operation in the context of mass customization and that Lean and Industry 4.0 can benefit from synergies because each one enhances the other. However, according to some authors [174], Industry 4.0 and Lean have essentially different approaches regarding the role digital technologies should have.

While some authors deem that TPS considers robots, machines and computers in the opposing side of *jidoka* ("automation with a human touch"), it should be noted that the lack of enthusiasm of TPS towards digital technologies could have been influenced by the current digital technologies of that era (the 1950s–1980s). Since the rate of change in digital technology has been particularly remarkable in the past four decades, it seems bold to assume that TPS's views on computers in the second half of the 20th century still apply.

### 3.3.9. Lean Tools for the Industry 4.0 Era

The arrival of the 4th industrial revolution could mean changes in the role or the value of existing Lean production tools. For example, Value Stream Mapping (VSM) could no longer be a sustainable tool since it might lack flexibility when dealing with digital processes, although evolutionary improvements to this tool could correct this shortcoming [179]. On the other hand, Lean automation aims at achieving the best possible combination of Lean and Industry 4.0 automation [180]. Industry 4.0 will create new forms of waste, digital waste, and Romero et al. conclude that future research would need to focus on new techniques developed to eliminate it [181,182]. Using simulations of Lean production environment can be used to find clustering alternatives that reduce the waiting time without compromising the business productivity [183]. Malik and Bilberg proposed a method for assigning tasks to robots or people in Human–Robot Collaborative (HRC) assembly, based on the physical properties of the components, HRC safety, and the dynamics of the HRC environment such as part presentation and feeding [184]. The IoT and simulation could be used to support expert-less decision making, in a similar way to the classic Andon tool does [185]. In any case, systems integration will be needed to ensure that Lean manufacturing systems meet the Industry 4.0 requirements [186].

In conclusion, classic Lean tools—e.g., value stream map—might need to change in order to remain useful for analyzing digital processes. The appearance of "digital waste" should be taken into account, but in general terms, Industry 4.0 technologies are expected to support the ability of people to make Lean-oriented decisions.

### 3.3.10. Lean Management Affected by the 4th Industrial Revolution

The evolution of Lean management in the context of Industry 4.0 leads to risks and opportunities. According to Rother et al. [187], the success factors of the coming transformation are three: management engagement, involvement and interaction. Therefore, the proposed approach is to use the technological advances to free up manager time and use it to focus on the human relationships: sharing knowledge, developing the workforce's skills and managing progress [188]. Total Quality Management will need to evolve as quality planning, quality control, quality assurance and quality improvement are different in a digital manufacturing framework compared to the previous human-capabilities-based era [189].

In conclusion, management has a key role to play in the successful transition to Industry 4.0. From the Lean perspective, changes brought by Industry 4.0 could be used to free up manager time to be invested focusing on human relationships.

### 3.3.11. Lean and Industry 4.0: Conclusions

Research on Lean tools for assembly operations is still an open topic. Firstly, it should be noted that since internal logistics are tightly associated with assembly, both should be analyzed together because changes to one will affect the other as well. Lean production systems typically employ assembly line or cell layouts to establish pull and create material flow. For certain contexts involving high-cost, high-variability, short product development cycles, *seru* assembly systems are particularly competitive because they are focused on adaptability. KPIs and performance assessment frameworks are used to measure the effects of changes in Lean production systems. Establishing a set of KPIs needs to take into account multiple stakeholders and to align the strategic and operational goals of the organization. Simulations and case studies show the beneficial effects of Lean methods and allow to estimate the economic return of investment of Lean management decisions.

The Toyota Production System (TPS) considers robots, machines and computers in the opposing side of *jidoka* ("automation with a human touch"), but it should be noted that their lack of enthusiasm towards digital technologies could have been influenced by the current digital technologies of that era (1950–80's). Since the rate of changes in digital technology has been particularly remarkable in the past four decades, it seems bold to assume that TPS's views on computers in the second half of the 20th century still apply. Currently, there is a wealth of evidence showing that Lean manufacturing is a valid approach to improve assembly operation in the context of mass customization and that Lean and Industry 4.0 can benefit from synergies because each one enhances the other. Some classic Lean tools—e.g., Value Stream Map—may need to change in order to remain useful for analyzing digital processes. In general terms, Industry 4.0 technologies are expected to support the ability of people to make Lean-oriented decisions. Management has a key role to play in the successful transition to Industry 4.0. From the Lean perspective, changes brought by Industry 4.0 could be used to free up manager time to be invested focusing on human relationships. Learning Factories could be a great tool to share the vision of Lean 4.0 assembly, but they need to mimic real-life scenarios to become useful for non-academic learners with industrial backgrounds, such as assembly operators. Industry 4.0 technologies could also be used to enhance the training environment of Learning Factories. Since both Lean and Industry 4.0 stress the importance of people, it seems only natural that supporting human capabilities becomes a priority in Lean 4.0 assembly

### 3.4. Focusing on People

In order to answer the fourth and last set of questions, "How would Industry 4.0 affect people in assembly?" and "How to support people transitioning to Assembly 4.0?", the systematic literature review publications related to the key concept "Operator" were analyzed. After a brief introduction, the six main topics to be considered will be presented, as shown in Figure 8: Line balancing, sequencing and job rotation; Lean: Operators at the center; Frameworks for operators in Industry 4.0; Automation and Human–Robot Collaboration; Supporting operators with Industry 4.0 technology; and Implications of smart factories for operators. Finally, the key conclusions will be summarized.

### 3.4.1. Introducing People in Assembly Operations

Human operators are critical for competitive assembly systems when considering information flows, competence needs and the requirements for effectively making use of automation. In such an environment, human teams—rather than individuals, are key [190]. The role of operators depends strongly on the type of production system (e.g., high-volume production vs. low-volume high-variety). Traditional automation allows increased productivity, but it lacks the adaptability of human operators. The design of reconfigurable assembly systems by incorporating both machines and people can lead to cost-effective system flexibility and scalability. However, the collaboration between people and robots can create safety issues. These can be addressed in two clearly separated ways, according to Hu et al.: (1) employing vision systems to stop robots; (2) robots so light and low force that they can be stopped

safely by people. Safely increasing flexibility and efficiency in mixed-model assembly lines is one of the problems that Industry 4.0 technologies seek to address [54].

**In conclusion**, the role of operators depends on the type of production system, and there is usually a trade-off between the increased productivity of automation and the adaptability of human operators. Reconfigurable, hybrid assembly systems that incorporate machines and people could lead to cost-effective flexibility and scalability. However, the collaboration between people and robots can also create safety issues.

**3.4 OPERATOR**

*Q4: How would Industry 4.0 affect people in Assembly?*
*How to support people transitioning to Assembly 4.0?*

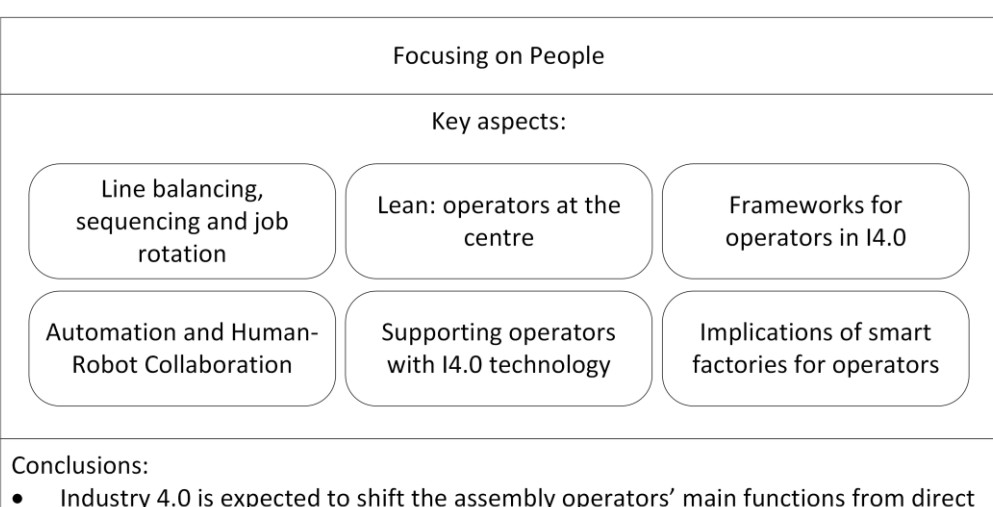

**Figure 8.** Key aspects of operators in Industry 4.0, and main conclusions of the analysis.

3.4.2. Line Balancing, Sequencing and Job Rotation

Having a flexible and cross-trained workforce is a recurrent approach to deal with the complexity and changing demand conditions of mass customization [191–193]. Operator job allocation can also be adjusted to address an array of situations: one-of-a-kind production [194], minimizing costs in *seru* production systems [195], high turnover and slow learning processes [196], a heterogeneous workforce with varying degrees of absenteeism [197], remarkable ergonomics and walking costs [198], or operator-intensive assembly optimization—along with sequencing [199]. Alternatively, sequencing algorithms can be used for minimizing operator headcount in reconfigurable assembly systems [200].

Although line reconfiguration is a common approach in mixed-model assembly, output can be increased in peak demand without it [201].

Analyzing the human operator characteristics and the process complexity can be used to maintain the process KPIs [202], to predict operator overload [203], or to assess human-originated quality problems [204]. Operator walking distances are a key input for kitting vs. line stocking decisions [205], and JIT kitting can be optimized by incorporating hybrid HRC systems [206].

**In conclusion**, a flexible and cross-trained workforce is key for dealing with changing demand conditions, allowing dynamic job assignation and efficient line balancing and sequencing.

### 3.4.3. Automation and Human-Robot Collaboration

Human-Robot Collaboration (HRC) expects to obtain the best of both human and automation worlds. Costa Mateus et al. developed a methodology for transitioning from manual to HRC assembly: (1) operation decomposition, (2) resource evaluation, (3) resource allocation, (4) collaborative assembly operation. [207]. However, HRC brings quality and reliability problems associated with robots and human operators separately, on top of their interactions, which needs to be addressed when establishing Quality Control [208]. Additionally, collaborative work with a robot has been found to cause stress in operators [209]. Moreover, operator safety remains a key concern for HRC systems. A safety strategy for HRC should consider the following key design areas: Human–Robot Collaboration spaces, robot safety systems, computer vision monitoring of safety conditions, and an operation control system that coordinates human–robot interaction [210]. Regarding the vision monitoring of safety conditions, Anton et al. used depth sensors so that robots avoid collisions with operators [211]. Another way of ensuring human operator safety in HRC would be the "safety bubble" concept, which is based on live data sharing between reconfigurable assembly systems [38].

**In conclusion**, Human–Robot Collaboration aims to obtain systems that are both flexible and highly productive. However, quality and safety concerns are yet to be solved.

### 3.4.4. Lean: Operators at the Centre

One key aspect of Lean production Systems (LPS) implementation is *respect for people*, which has been typically overseen [212]. Worker development defines the Toyota Production System (TPS) culture of respect and teamwork, and although it does not directly relate to bottom-line results, it is an integral component of the TPS implementation of *kaizen* (continuous improvement) [213]. There are simple ways to involve operators and supervisors in the continuous improvement journey, and they are built on showing the importance and effect of everyone's actions towards addressing the problems together [214]. One-point lessons have been found effective in sustaining the standardization and optimization in LPS [215].

There must be a balance between worker autonomy and creativity versus process and cost control, and De Haan et al. found that "challenging and enabling workers to creatively use their talent and skills in daily work will most likely lead to positive results" [216]. Another tension exists related to judgment-based operator adjustments to processes, which could be considered as tampering from the Statistical Process Control (SPC) point of view. Operator adjustment is not always bad, but a necessity in real production plants, and there are methods to determine whether the operator judgment was appropriate or not [217].

Romero et al. looked towards *Jidoka* (or "automation with a human touch") when analyzing the future relationship of people and machines in the emerging 4th Industrial Revolution. They stress that *Jidoka* needs to be understood not only as an approach to automation but also as a "learning system" in which machine and human benefit from each other [4]. "Employee development system", a tool of Lean production management, can be used to enhance the problem-solving capabilities of the workforce, which leads to improved results measured by KPIs [218].

New frameworks consider people as the cornerstone of LPS: either depicting them as one of the fundamental pillars—alongside processes and tools [219]; or directly as the center of a layered model for Lean factory design [220].

**In conclusion**, "respect for people"—a core principle of Lean production—should be considered a cornerstone of Lean production process design. There must be a balance between worker's autonomy and process control, keeping in mind that operators' involvement in the continuous improvement journey is necessary for success in the long term.

### 3.4.5. Frameworks for Operators in Industry 4.0

The concept of Industry 4.0 appeared to provide cohesion to different visions regarding the future of manufacturing, connected by Key Enabling Technologies (KET). Alongside the development of such technologies, recent research has focused on theoretical frameworks to conceptualize the use of the KET and its impact on human operators. Lindblom et al. [221] studied how to evaluate the Human–Robot Collaboration in terms of safety, trust and operator experience; Golan et al. [222] looked into the future Industry 4.0 interaction between operator and workstation, composed of three subsystems: observation, analysis and reaction.

The key role of operators in the era of the 4th Industrial Revolution has been identified by numerous authors, coining the term *Operator 4.0* [223]. Industry 4.0 technologies should support operators in their tasks, either by directly helping them or by providing meaningful information to assembly system design engineers. Peruzzini et al. developed a theoretical human-centered framework for Operator 4.0 using digital twin-based simulation, and real-time human data capture can be used to provide insights on operator ergonomics and mental workload [69]. In a similar way, Mattson et al. propose a method of measuring the wellbeing and performance of operators at assembly stations using electro-dermal activity [88]. Industrial IoT is another technology that can be used for capturing human and machinery data for understanding human impact on Industry 4.0 assembly systems [65]. Understanding the operator's information needs is vital for the design of smart assembly factories [57].

**In conclusion**, new operator-centered frameworks are appearing to conceptualize the role of people in the 4th Industrial Revolution era. The key role of operators has been identified by numerous authors, coining the term operator 4.0 [223]. Industry 4.0 technologies should support operators in their tasks, either by directly helping them or by providing meaningful information to assembly system design engineers.

### 3.4.6. Supporting Operators with Industry 4.0 Technologies

Industry 4.0 technologies offer new ways to support human operators in their duties—see Table 3: training can be made easier with Virtual Reality (VR), Augmented Reality (AR) and motion tracking [74–76]; instructions can be generated in real time and displayed using AR [77–79]; or projection AR can be used to provide process information [80], assembly assistance [81], safety in HRC "chaotic" smart warehouses [224], shipyard worker assistance [225] or to enhance the operator's capabilities and competencies [82]. In general, human operators are positive about the use of AR for assembly support [226]. The technology-enhanced operator is a growing field of research, with many other Industry 4.0 KET involved in achieving varied goals: IoT-based human–Cyber Physical Systems for providing feedback to operators working in an intelligent space [72]; reducing big data to smart data to assist people [60]; software robots (softbots) to interface between machines and computer information systems [227]; mobile devices in order to allow dynamic job rotation in multi-variant assembly lines [228]; verbal and visual prompts for assisting workers with intellectual disabilities [229]; wearables for audio commands [230] or detecting potentially hazardous or risky situations [231]; or a combination of many technology-enabled tools [232–234].

**In conclusion**, varied Industry 4.0's Key Enabling Technologies can be used to support production operators to obtain different benefits. In particular, Virtual and augmented reality and wearable devices have attracted great attention. Operators can be supported with assembly instructions, quality control,

assembly details prompts or enhanced training programs, which can be provided in a way that is satisfactory for the users.

### 3.4.7. Implications of Smart Factories for Human Operators

Digital technologies' progressive presence in factories will change the role of human operators, which will shift from work-focused activities towards dispositive tasks, supervision and decision activities [235]. Operators will, therefore, need more information than ever before, and these requirements need to be carefully assessed [57]. Considering the operator at the center, human activities with Cyber Physical Systems (CPS) have been modeled, and new KPIs proposed to make visible how business and operational decisions affect operators [113]. Empowering operators seems one possible way of making Smart factories happen, and such empowerment will make visual computing technologies necessary, according to Segura et al. [236].

Digital technologies can also be used to obtain insights into human–machine interactions [65] or worker's wellbeing [88], which then lead to forming strategies for cognitive automation [49]. Despite recent advances, digital maturity in manufacturing companies has a long way to go, and most operator-machine interaction is done by mouse and keyboard hardware instead of by using CPS [237].

**In conclusion**, human operators will need to receive and manage more information than ever before, make decisions and supervise instead of focusing on mechanical work-related activities. Therefore, empowering operators to act more autonomously and supporting them accordingly seems necessary. To understand the situation of Industry 4.0 operators can be done using new digital technologies, obtaining meaningful data in ways that were not possible before.

### 3.4.8. Focusing on People: Conclusions

The role of operators depends on the type of production system, and there is usually a trade-off between the increased productivity of automation and the adaptability of human operators. Reconfigurable, hybrid assembly systems that incorporate machines and people could lead to cost-effective flexibility and scalability. However, the collaboration between people and robots can also create safety issues. There must be a balance between worker's autonomy and process control, keeping in mind that operators' involvement in the continuous improvement journey is necessary for success in the long term. "Respect for people"—a core principle of Lean production—should be considered a cornerstone of Lean production process design. A flexible and cross-trained workforce is key for dealing with changing demand conditions, allowing dynamic job assignation and efficient line balancing and sequencing. New operator-centered frameworks are appearing to conceptualize the role of people in the 4th Industrial Revolution era. The key role of operators has been identified by numerous authors, coining the term operator 4.0. Industry 4.0's Key Enabling Technologies can be used to support production operators to obtain different benefits. In particular, Virtual and Augmented Reality and wearable devices have attracted great attention. Operators can be supported with assembly instructions, quality control, assembly details prompts or enhanced training programs, which can be provided in a way that is satisfactory for the users. Human operators will need to receive and manage more information than ever before, make decisions and supervise instead of focusing on mechanical work-related activities. Therefore, empowering operators to act more autonomously and supporting them accordingly seems necessary.

## 4. Discussion

This section outlines the key ideas of the four areas considered in the previous section, organized as answers to the four sets of questions posed in the introduction.

### 4.1. Assembly & Mass Customisation

The question related to Assembly and Mass customization is: "What are the characteristics and implications of mass customization for assembly operations? "

Mass customization brings increased complexity that needs to be addressed at multiple levels and taking a holistic point of view to ensure that optimizing a subsystem does not negatively affect another subsystem. Assembly complexity reaches outside the boundaries of assembly operations and needs to be considered jointly with supply chain, quality, maintenance and IT/IS. Industry 4.0 digital technologies have a critical role to play in making possible mass customization assembly systems that do not compromise on quality and cost.

### 4.2. Industry 4.0 & Key Performance Indicators

The set of questions related to Industry 4.0 and KPIs are: "What new Industry 4.0 digital technologies are relevant to assembly operations?", "How to measure the improvement?" and "How to make the most out of them?"

There are many examples of new technologies applied to final assembly—see Table 3: the Internet of Things, big data and digital twins for improving processes and decisions as well as for gathering data and obtaining valuable information; Cyber Physical Systems and Augmented/Virtual Reality for measuring human operator performance and supporting human operators' work; and a mix of technologies to support different aspects that enable mass customization. However, assembly operations are likely to experiment an evolution rather than a revolution by gradually incorporating these technologies. Two key areas will be of particular interest: enhancing the role of people in assembly operations—especially in terms of responsibility and skills; and making possible human–machine hybrid systems capable of efficient low-volume high-variability production.

To evaluate the performance of assembly systems, a KPI system is employed. Standard KPIs need to be adapted in order to include both traditional metrics (e.g., cost, quality, throughput, inventory, lead time, productivity) and new indicators that are specific to the products, operations, stakeholders, business goals and IT-related aspects of the smart factory.

Despite the wealth in the literature about what new technology can do, few relate to methodologies to assess the operational needs and opportunities in final assembly and then finding or developing an Industry 4.0 solution to them.

### 4.3. Lean Assembly for Industry 4.0

The question related to Lean production is: "Is Lean production the best starting ground for implementing Industry 4.0 assembly operations?"

Lean manufacturing offers an array of tools and techniques to deal with the increasing demand complexity and variability, and which could benefit assembly operations in the context of mass customization. While most authors consider Lean manufacturing as a valid approach for increased complexity of mass customization, others claim that Lean cannot be applied straightforwardly in the Industry 4.0 era. Lean might not be necessarily the best possible starting ground for smart assembly in every situation. However, it clearly has positive synergy with Industry 4.0 because new technologies can enhance Lean assembly, and Lean maturity supports the implementation of new technology. Moreover, both Industry 4.0 and Lean consider that people have a central role to play in assembly operations.

### 4.4. Assembly Operators in Industry 4.0

The questions related to human operators are: "How would Industry 4.0 affect people in assembly?" and "How to support people transitioning to Assembly 4.0?"

Industry 4.0 is expected to shift the assembly operators' main functions from direct labor activities to managing information and making decisions, supported by technology. A flexible and cross-trained workforce would be key for dealing with changing demand conditions, allowing dynamic job assignation, line balancing and sequencing. Learning factories are a great way to train operators in the new digital manufacturing skills needed for smart factories and to gain a deeper understanding of how new technologies affect them.

## 5. Conclusions

This article looked at the issue of how Industry 4.0 technologies could improve the flexibility, productivity and quality of assembly operations. To do so, a systematic literature review was carried out, and 239 articles were analyzed. The resulting analysis was structured into four main topics, each one addressing one of the questions posed in the introduction.

It was found that mass customization brings complexity into assembly operations, which need to be looked at from a holistic point of view—joining assembly, supply chain, quality, maintenance and IT. New technologies—such as big data, the Internet of Things, real-time optimization, cloud computing, CPS, Virtual/Augmented Reality, additive manufacturing and digital twins–allow obtaining meaningful information in real time about the assembly operations, making better decisions and supporting human operators in their activities. A combination of conventional and new KPIs to evaluate IT-related aspects of the smart factory will be needed to measure the impact of these technologies. Although it might not necessarily be the best starting point in each and every situation, Lean is definitely a great starting ground for smart factories. Since both Industry 4.0 and Lean consider that people have a critical role to play in assembly operations, frameworks that place human operators at the center of Lean 4.0 have started to appear. This focus will need to be translated into supporting people to acquire the digital manufacturing skills they will need. Learning Factories are great to this end.

The literature analysis also uncovered the relative lack of methodologies for implementing Industry 4.0 technologies in assembly operations to address concrete business goals, which remains an open question. There is also room for developing operator-centered frameworks for Industry 4.0 that are specific to assembly operations in the demand context of mass customization.

**Author Contributions:** Conceptualization, M.T., J.A.Y.-F. and A.M.; methodology, A.M.; investigation, A.M.; writing—original draft preparation, A.M.; writing—review and editing, M.T., J.A.Y.-F. and A.M.; visualization, A.M.; supervision, M.T. and J.A.Y.-F. All authors have read and agreed to the published version of the manuscript.

**Funding:** This project has received funding from the European Union's H2020 research and innovation programme under the Marie Sklodowska-Curie Actions. Grant Agreement no. 814225.

**Conflicts of Interest:** The authors declare no conflict of interest.

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
