# Peer review of "Lean Manual Assembly 4.0: A Systematic Review"

_applsci, doi:10.3390/app10238555_

Round 1
Reviewer 1 Report
The topic is actual and common field of work nowadays. The paper has a logical structure and is clearly, concisely and accuratel written. However the results are to general in some respect: “Lean is a great starting point for implementing such changes” .
Why did the authors set the time window from 2010-2020?
Why did not the authors deal with the following keywords: TPS, KAIZEN, Cell capacity?
It would be great if a list of the Industry 4.0 technologies and a list of the KPIs added to the conclusion part.
Comments:
Line 429. …”traditional and new IT-related KPIs..”; It would be great to see a list about the KPIs.
Line 518 … “research on Lean tools for assembly operations is still an open topic” How did the authors draw this conclusion? The sources from 124-133 mostly case studies. We can see that how a company used the technique (VSM, or Standardized Work, etc.). I would say that the industry uses the LEAN techniques and tools based on the sources 124-133. How can you prove that: there are existing research in that field, based on the source 124-133?
Line 644-645 … “It is clear that TPS considers robots, machines and computers in the opposing side of jidoka (‘automation with a human touch’)” Explain it in detail. What does opposing side means?
Line 656 …” new techniques are” Could the authors write examples?
Line 907. Could the authors specify the cost and quality KPIs. Cost of what? (activity, process, manufacturing, quality etc.. ) What are the available quality KPIs? (ppm, FTQ – First time Quality etc.. )
Fig 6. Conclusion:
- 1st point: It would be great to see a list about the tools (as a summary)
- 2nd point: Could you determine the percentage of the different points of view?
- 3rd point: Can you state that: One of the most important part of Industry 4.0 is LEAN. (that is the reason of the synergy.) ? – Can Industry 4.0. exist without LEAN?
Reviewer 2 Report
In this review paper authors looked at the issue of how Industry 4.0 technologies could improve the flexibility, productivity and quality of assembly operations. A systematic literature review was carried out, and 239 articles were analysed.
This is an important and potentially interesting topic and the paper is generally well-written. The theme is topical and may interest a wide audience.
The authors have clearly done a lot of work, but their research design is not set out very clearly. They have set out some interesting hypotheses, leading to some tentative conclusions.
My main problem with this paper lies in the arguing the concepts used; I expect all terms to be coherently explained using comparisons with other studies in the field.
For example, please explain why you choose this 6 key concepts:
The 6 key concepts that were used are: Assembly, Mass Customisation, Key Performance Indicator (KPI), Lean Manufacturing, Industry 4.0 and Operator.
The authors use many key concepts that are not explained very clearly, nor why they were used in literature research. For a better understanding of the paper, I recommend some schemes that can highlight the connection between concepts.
